

# The effects of hip- and ankle-focused exercise intervention on dynamic knee valgus: a systematic review

Farhah Nadhirah Aiman Sahabuddin[1], Nazatul Izzati Jamaludin[1], Nurul Hidayah Amir[2,3] and Shazlin Shaharudin[1]

[1] Exercise and Sports Science Programme, School of Health Sciences, Universiti Sains Malaysia, Kota Bharu, Kelantan, Malaysia
[2] Department of Translational Health Sciences, Faculty of Health Science, University of Bristol, Bristol, United Kingdom
[3] Faculty of Sports Science and Recreation, Universiti Teknologi MARA, Arau, Perlis, Malaysia

## ABSTRACT

**Background**. A range of non-contact injuries such as anterior cruciate ligament tear, and patellofemoral pain syndrome are caused by disordered knee joint loading from excessive dynamic knee valgus (DKV). Previous systematic reviews showed that DKV could be modified through the influence of hip strength and ankle range of motion. Therefore, the purpose of this systematic review was to examine the effects of exercise intervention which involved either top-down or bottom-up kinetic chains on minimizing DKV in male and female adults and adolescents, with and without existing knee pain.

**Methodology**. Electronic searches were conducted in SAGE, Science Direct, SCOPUS, and Pubmed. The search strategy consisted of medical subject headings and free-text search keywords, synonyms and variations of 'exercise intervention,' 'knee alignment,' 'dynamic knee valgus', 'knee abduction' that were merged via the Boolean operator 'AND' and 'OR'. The search was conducted on full-text journals that documented the impact of the exercise intervention program involving either the bottom-up or top-down DKV mechanism on the knee kinematics. Furthermore, exercise intervention in this review should last at least one week which included two or three sessions per week. This review also considered both men and women of all ages with a healthy or symptomatic knee problem. The risk of bias of the included studies was assessed by Cochrane risk assessment tool. The protocol of this review was registered at PROSPERO (registration number: CRD42021219121).

**Results**. Ten studies with a total of 423 participants (male = 22.7%, female = 77.3%; adults = 249, adolescents = 123; pre-adolescent = 51) met the inclusion criteria of this review. Seven studies showed the significant effects of the exercise intervention program (range from two weeks to ten weeks) on reducing DKV. The exercise training in these seven studies focused on muscle groups directly attached to the knee joint such as hamstrings and gastrocnemius. The remaining three studies did not show significant improvement in DKV after the exercise intervention (range between eight weeks to twelve weeks) probably because they focused on trunk and back muscles instead of muscles crossing the knee joint.

**Conclusion**. Exercises targeting specific knee-joint muscles, either from top-down or bottom-up kinetic chain, are likely to reduce DKV formation. These results may assist

Corresponding author
Shazlin Shaharudin, shazlin@usm.my

athletes and coaches to develop effective exercise program that could minimize DKV and ultimately prevent lower limb injuries.

# INTRODUCTION

Dynamic knee valgus (DKV) is defined as a body position in which the knee collapses from excessive valgus, excessive internal-external rotation, or both conditions (*Krosshaug et al., 2007*). DKV can be caused by hip abductor weakness that entails internal rotation of the hip, excessive frontal knee alignment or tibial rotation angles and contralateral pelvic drop (*Powers, 2010*). Disordered knee joint loading from excessive DKV can trigger a spectrum of injuries such as anterior cruciate ligament (ACL) tear, and patellofemoral pain syndrome (PFPS) (*Myer et al., 2015*).

Two types of kinetic chain play a major role in the mechanism of DKV, namely top-down (proximal origin) and bottom-up (distal origin) kinetic chains (*Jamaludin et al., 2020*). A top-down kinetic chain occurs when the hip and trunk muscles alter the distal joints' kinematic patterns (*Snyder et al., 2009*). On the contrary, a bottom-up kinetic chain involves the influence of ankle musculature and foot structures on knee joint motions (*Khamis & Yizhar, 2007*). Several studies have reported that the strength of hip adduction, knee flexion, and knee extension were the key indicators of knee valgus (top-down kinetic chain) (*Willson et al., 2011*; *Willson, Ireland & Davis, 2006*). Besides, recent studies also observed that excessive DKV could be associated with foot-ankle strength as well as its range of motion (ROM) and kinematics (bottom-up kinetic chain) (*Kagaya, Fujii & Nishizono, 2015*; *Lima et al., 2018*). Previous studies aimed to find the sources of excessive DKV and knee joint loading to prevent lower limb injuries such as ACL strain (*Nessler, Denney & Sampley, 2017*). Several other studies also presented evidence on how preventive training programs could minimize the occurrence of non-contact lower limb (*Petersen et al., 2005*), knee or ACL (*Hewett et al., 1999*), and ankle injuries (*Verhagen et al., 2004*).

A systematic review by *Lima et al. (2018)* evaluated the association between ankle dorsiflexion (bottom-up kinetic chain) and DKV in interventional and non-interventional studies. By including the non-interventional studies, the review results might not be related to the impact of exercises on reducing DKV. Meanwhile, a systematic review by *Dix et al. (2018)* investigated the association between hip muscle strength (top-down kinetic chain) and DKV in asymptomatic (i.e., free from any injury) females. Similarly, the review did not investigate the effects of exercises that contribute to hip muscle strength in reducing DKV. A narrative review by *Ford et al. (2015)* provided details on a hip-focused neuromuscular exercise intervention to improve DKV. However, the narrative review by *Ford et al. (2015)* excluded in-depth and systematic literature search approach; hence, it may miss out several relevant papers. The review with meta-analysis conducted by *Lopes et al. (2018)* was focused on the effects of injury prevention programs (IPPs) specifically on landing

biomechanics as they relate to the ligament, quadriceps, trunk, and leg dominance theories of ACL injury. To the best of our knowledge, no systematic review has been conducted that focused on the effects exercise intervention based on top-down (hip-focused) or bottom-up (ankle-focused) kinetic chain on DKV. The review will shed light on how the exercise training programs may improve the mechanisms behind knee injury. Therefore, the present systematic review aims to determine the influence of hip- and ankle-focused exercise intervention on improving DKV.

## METHODS

This review was conducted in compliance with the Preferred Reporting Items for Systematic Reviews (PRISMA) (*Liberati et al., 2009*). Universiti Sains Malaysia granted ethical approval to carry out the study within its facilities (Ethical Application Ref: USM/JEPeM/18070316). The protocol of this review was registered at PROSPERO (registration number: CRD42021219121).

### Search Strategy

Two researchers (FNAS and NIJ) individually screened through four medical databases, namely SCOPUS, SAGE, Pubmed, and Science Direct from database inception until November 2020. The search technique consisted of medical subject headings (MeSH) and free text search keywords, synonyms, and variations to retrieve all relevant articles. Three phrases were merged for searching databases using the Boolean operator 'AND' and 'OR': i.e., 'exercise intervention,' 'training', 'knee alignment,' 'dynamic knee valgus', 'knee abduction'. The reference lists of all the included manuscripts and authors' files were also reviewed to identify other relevant studies.

### Study selection

The titles and abstracts of the retrieved studies were downloaded into Mendeley (version 1.19.4; Mendeley, London, United Kingdom). Two independent reviewers scanned all abstracts for eligibility and removed any duplicates. Full texts were obtained for abstracts that fulfilled the inclusion criteria. In the event of any ambiguous details, the corresponding authors of the studies were contacted via e-mail. Any disagreement between the two investigators would be resolved by discussing with the third investigator (SS) so that a consensus could be reached. A schematic diagram of the study selection is shown in Fig. 1.

### Inclusion and exclusion criteria

To be eligible for inclusion, the studies must describe the impact of the exercise intervention program that involved either the bottom-up or the top-down DKV mechanism on the knee kinematics. The duration of the intervention must be at least one week with two or three sessions per week. Men and women of all ages with healthy or symptomatic knee conditions were included. There was no limitation on the date of publication. Only human interventional studies presented in English full-text journals that discussed DKV or knee alignment were included in this review. Other research designs such as meta-analyses, systematic reviews, case reports and series, cross-sectional studies, concept papers,
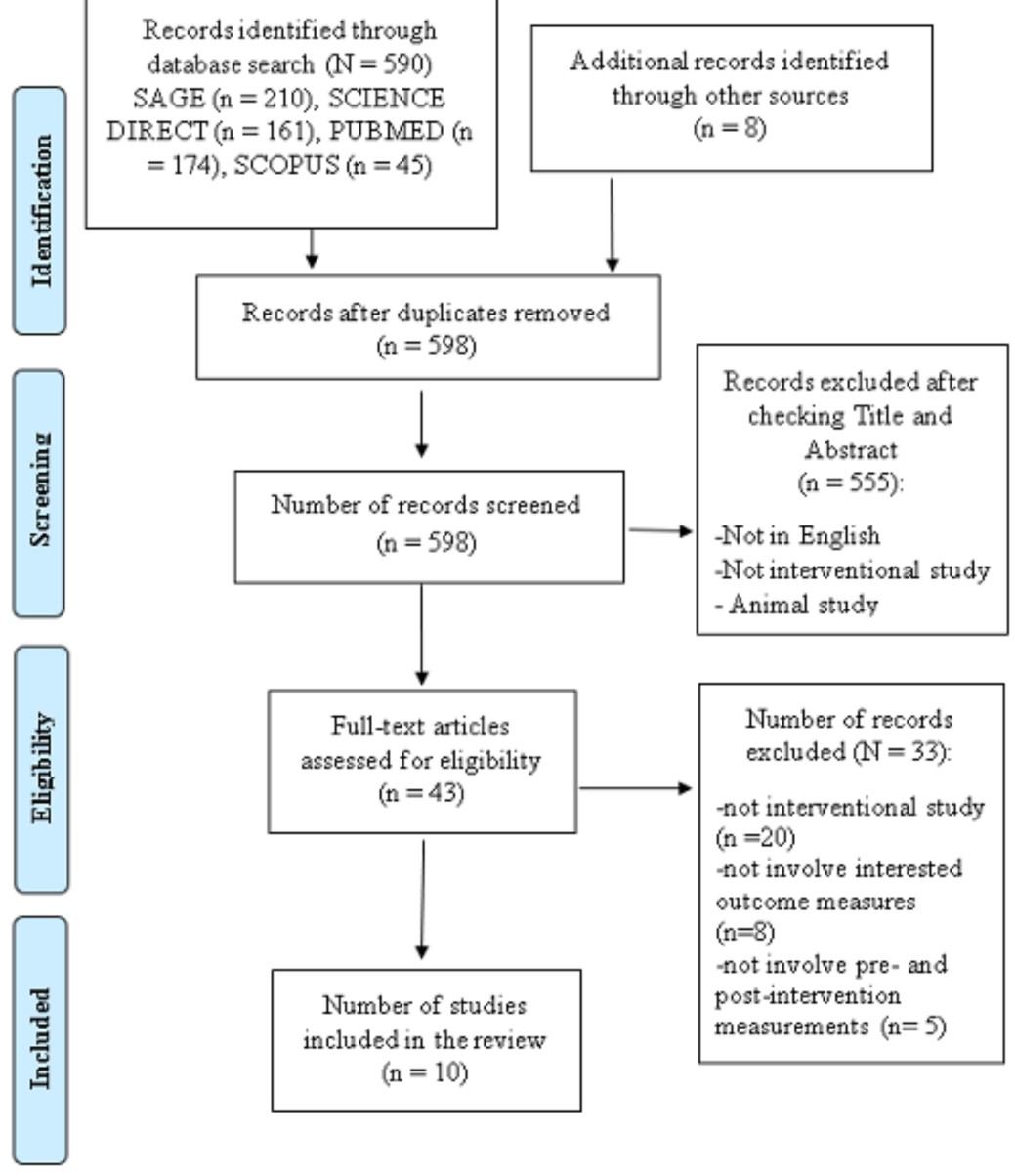

**Figure 1** PRISMA (Preferred Reporting Items for Systematic Reviews and Meta Analyses) flow chart of the included studies.

editorials, opinions, and in vitro research were excluded. Seminar, poster presentations, reviews, case studies, editorials, letters, and abstract-only texts were also excluded.

## Data extraction

For every included study, two researchers extracted the information. The data were synthesized and tabulated based on the first authors' surname, date of publication, sample size, participants' age and fitness level, exercise intervention and its duration, group allocations, methods of outcome assessment, functional tasks, and outcomes (Table 1).

Sahabuddin et al. (2021), PeerJ, DOI 10.7717/peerj.11731

**Table 1 Characteristics of included studies.**

| Study | Participant's characteristics | Program duration, sessions per week | Groups | | Task | Outcome measures |
|---|---|---|---|---|---|---|
| *Sheerin, Hume & Whatman (2012)* | $n = 19$ (11 male, 8 female), 9 to 14 years old (11.54 ± 1.34 years) Healthy youth athletes, competitive sports. Presence of DKV was determined based on knee abduction angle from pre-intervention | -8 weeks -3 times per week | Control group ($n = 10$): Open and closed kinetic chain upper limb strengthening exercises: -low pulley row and over-head pull-down with a resistance band, -bicep curls -lying chest press -front and side shoulder raises -overhead press with small hand-weights -triceps dips from a bench. | Experimental group ($n = 9$) Similar training as control with additional functional weight bearing exercises:- -side lying hip abduction -double leg squats -crab walking - standing hip abduction -single leg squat -jump squats -jumps squats with rotation 90° and 180° -Broad jump (forward deep hold, single leg) -double leg landing | Treadmill-based assessment of running gait with 3D analysis (while wearing shoes) | Differences in pre- to post-intervention changes between control and experimental groups: -trivial for the right knee (−0.3°) -large detrimental increase in left knee valgus angle (1.9°) |
| *Baldon et al. (2014)* | $n = 31$ females 18-30 years old (ST = 22.7 ± 3.2, FST = 21.3 ± 2.6) Diagnosed with Patellofemoral Pain (PFP), recreational athletes (athletic activity for at least 3 times per week) DKV status was based on diagnosed of PFP. | -8 weeks -3 times per week Duration: ST = 75-90 min per week FST= 90-120 min per week | ST ($n = 16$) Quadriceps and lateral retinaculum, hamstrings, soleus, gastrocnemius, and iliotibial band stretches exercise -straight leg raise in supine -seated knee extension (90° -45 ° of knee flexion) -leg press (0° -45° of knee flexion) -wall squat (0° -60 ° of knee flexion) -step-ups and step-downs from a 20-cm step -single leg standing on unstable platform. | FST ($n = 15$) Transversus abdominis and multifidus muscle training exercise -lateral and ventral bridge -trunk extension on Swiss ball -isometric hip abduction/ lateral rotation in standing - hip abduction/ lateral rotation/ extension in side-lying - hip extension/ lateral rotation in prone - hip abduction/ lateral rotation with slight knee and hip flexion in side-lying - pelvic drop during standing - hip lateral rotation in closed kinetic chain - single-leg deadlift - single-leg squat - forward lunge - prone knee flexion - seated knee extension (90° -45° of knee flexion) -single-leg standing on unstable platform. | Single Leg Squat test with at least 60° of knee flexion | -significant reduction of knee abduction moment after 8-week of intervention in FST only. |

Sahabuddin et al. (2021), *PeerJ*, DOI 10.7717/peerj.11731

**Table 1** (*continued*)

| Study | Participant's characteristics | Program duration, sessions per week | Groups | | Task | Outcome measures |
|---|---|---|---|---|---|---|
| | | | The strengthening exercises performed by both groups were based on 1-RM, pain not more than 3/10 of 10-cm VAS scale. Loads were progressed across intervention when patients can perform without excessive knee pain, fatigue and muscle pain 48 h after training session. | | | |
| *Barendrecht et al. (2011)* | $N = 80$ Age (years): -AAVA (NMT = (15.6 ± 1.5) & RT = (15.6 ± 1.5)) -BAVA (NMT = (14.9 ± 1.3) & RT = (15.2 ± 1.3)), Handball player (males = 34; females = 46) The status of DKV was divided into two groups (AAVA and BAVA) based on drop-jump test. | -10 weeks -2 sessions per week | NMT AAVA ($n = 27$) BAVA ($n = 22$) -usual handball training and standard warm-up. -balance and coordination exercises on a wobble board and a mat -strength and plyometric exercises | RT AAVA ($n = 22$) BAVA ($n = 9$) -performed standard warm- up and the usual handball training only. | Test: Drop jump test (2D) from 30-cm height with double leg landing. The highest jump of 2 trials was measured. | Linear regression analysis showed that in the NMT groups (AAVA and BAVA groups), initial minimum normalized knee distance predicted 52% of the variance in pre- to post-test for knee flexion angle. |
| *Bell et al. (2013)* | $N = 32$ Volunteers (males:3) Age, (years) = (control group = (20.4 ± 2.9), Intervention group = (20.9 ± 2.6), DKV was screened in double-leg squats test. | -3 weeks -2 or 3 sessions per week | Control Group ($n = 16$) -no exercise intervention -only perform functional test at pre- and post-intervention. | Intervention Group ($n = 16$) -single limb balance with squat (1–3 set × 10–15 rep) -single limb balance with squat on unstable surface (2–3 set x 10 rep) -star excursion balance (2 to 3 set x 10 rep) -star excursion balance on unstable surface (2 to 3 set x 10 rep) -hop to balance (3 set x 10 rep) | Test: Double-legged squat at 90° of knee flexion | Only the intervention group showed a significant improvement on the knee valgus by 5° (64% reduction) during half squat. |
| *Czasche et al. (2017)* | $N = 16$, Age (years) = Control group = (22.9 ± 2.4) & Intervention group = (22.0 ± 3.2) Untrained female took part in recreational physical activity at most 4 times per week. DKV was not screened. | -8 weeks -180 min sessions per week | Control group ($n = 8$) -continue their routine recreational activities | Intervention group ($n = 8$) performed strength training exercise: -split squat / bulgarian -lunge -step-up -single leg bridge -squats -single leg good morning -single leg hip thrust -stiff leg deadlift *Loads were progressively increased across 8 weeks based on individual responses to training (strength, experience and motivation) -sets, reps, rest and perceived exertion were fixed. | Test: Bilateral landings (BL) and unilateral landings (UL) both from 30 cm platform. | Kinematics in both BL and UL in intervention group showed no significant differences from pre- to post-test. |

Sahabuddin et al. (2021), *PeerJ*, DOI 10.7717/peerj.11731

**Table 1** (*continued*)

| Study | Participant's characteristics | Program duration, sessions per week | Groups | | | | Task | Outcome measures |
|---|---|---|---|---|---|---|---|---|
| *Saad et al. (2018)* | $N = 40$, Age (years)= Quadriceps ($n = 23.2 \pm 2.53$), HIP ($22.5 \pm 1.08$), Stretching ($21.3 \pm 1.16$) and Control ($23.2 \pm 1.03$)) Women with PFP, Participate in aerobic or athletic activity at least 3 times per week at least 30 min. DKV status was based on diagnosed of PFP. | -8 weeks -2 sessions per week | QG ($n = 10$) - quadriceps strengthening exercises. | HG ($n = 10$) - hip strengthening exercises | SG ($n = 10$) -physical therapists monitored and stabilized the patients while performing stretching exercises for all muscles involved in knee and hip stabilization. | CG ($n = 10$) -N treatment | Test: Step-up and down -Dynamic valgus were measured in frontal plane at 45° tibiofemoral flexion during step-down task. | At pre-test, participants showed valgus movement (87.18% during step-up, and 82.05% during step-down) These values declined at the end of intervention around 66.67% and 48.72% respectively in HG and QG only. |
| *Araújo et al. (2017)* | $N = 34$ Age (years): (Experimental group $= 22.41 \pm 3.81$; Control group $= 21.71 \pm 2.08$) Female university students. DKV status: present in both groups. Screened DKV during assessment of step-down task. | -8 weeks -3 session per week | Control Group ($n = 17$) -continued their usual routines activities. | | Experimental Group ($n = 17$) -strengthening exercises: -hip lateral rotators -gluteus medium latissimus dorsi abdominal oblique and quadratus lumborum -lateral rotation and extension of the hip and trunk in closed kinematic chain. 1st to 2nd weeks –minimal load 3rd to 4th weeks –70–80% of 1RM 5th to 8th weeks –90–100% of 1RM Three sets of eight repetitions with a one-minute interval between sets. *If the participant performed three sets of nine repetitions on two consecutive sessions, she would perform the exercise with the load increased by 10% or 5%, in the consecutive session. | | Test: Step-down from 18cm step | Knee kinematics in transverse plane during step-down was not statistically significant after intervention in both groups. |
| *Thompson-Kolesar et al. (2017)* | $N = 94$ Age (years): Preadolescent (intervention: $11.8 \pm 0.8$; control: $11.2 \pm 0.6$) Adolescent (intervention: $15.9 \pm 0.9$; control: $15.7 \pm 1.1$). Female soccer athlete. DKV status: Preadolescent displayed greater knee valgus than adolescent athlete. Screened DKV from the baseline for all tasks. | 15 sessions -2 sessions per week -7–8 weeks | Control group: (Preadolescent ($n = 23$) & adolescent athlete ($n = 21$)) Continued their daily routine. | | Intervention group: (Preadolescent ($n = 28$) & adolescent athlete ($n = 22$)) Each session lasted about 25 min and served as a replacement for each team's standard warm-up prior to daily practice. Exercises for core strength, neuromuscular control, balance, eccentric hamstring training, plyometric, and agility were included. | | Test: -Single, double-legged jump from 30 cm box. -Preplanned and unanticipated cutting. *running sidestepping cut off the dominant limb at 45° from the line approach. | A significant group effect in the pre- to post change for initial touch knee valgus angle ($P = .004$) and peak knee valgus moment ($P = .011$) during the double-legged jump test throughout the intervention group. Changes in initial contact and peak knee valgus angles were not significantly different for all activities in both age groups. |

| Study | Participant's characteristics | Program duration, sessions per week | Groups | | Task | Outcome measures |
|---|---|---|---|---|---|---|
| *Jeong, Choi & Shin (2020)* | N = 48, Age (years) = Control group = (23.1 ± 1.2) & Intervention group = (22.4 ± 2.6) Recreationally active males. DKV was not screened. | 10 weeks -2/3 sessions per week | Control group (n = 16) Continued their usual routine. | Intervention group (n = 32) Warm-up, 1.3-km jogging Core training -Leg raise -Crunch -Superman -Plank hip twist -Prone-plank -Side-plank (both sides) -Supine bridge Stretches -Quadriceps -Hamstring -Calf stretches -Latissimus dorsi -Hip muscles -Pectorals/biceps | Side step cutting: -Run 3 m at a speed of 3.5 to 4.0 m/s before contacting their dominant foot on the force plate and then switching to the opposite limb. The line was marked on the floor to cut at 45° angle in the direction of progression. | After training, the knee valgus angles at initial contact significantly decreased by 46%, compared with the corresponding values in pre training (P =.038) No significant differences were observed in the knee valgus for control group. |
| *McCurdy et al. (2012)* | N = 29, Age (years) = (21.04 ± 1.83) Female with previous high school athletic experience. DKV was not screened. | 8 weeks -2 sessions per week | Control group (n = 16) -Continued their routine as usual. | Experimental group (n = 13) -Bilateral squat -Lunge -Step up -Romanian deadlift -Unilateral squat -A linear periodized program was applied by increasing 5% each week while the volume decreased. -Progression was adopted by including bilateral squat and Romanian deadlift in the first weeks of 8-week period while unilateral exercises were added in the mid-training period. -The intensity:- Bilateral squat- 50%–85% of 1RM Other exercises- 6-15RM | Drop jump task: -Jump from 60 cm for bilateral jumps -Jump from 30 cm box for unilateral jumps | No significant main effects for training groups (control vs. resistance trained) or trials (pretest vs. posttest). No significant difference between pre- and post-test knee flexion values in the control group for the unilateral jump, but there was a significant decrease from pre-test (82.4 ± 3.9) to post-test (69.6 ± 5.2) values for the bilateral jump. |

**Notes.**

Abbreviation: DKV, Dynamic knee valgus; FST, Functional Stabilization Training; ST, Standard Training; PFP, Patellofemoral pain; AAVA, Above-Average Valgus Aligned; BAVA, Below Average Valgus Aligned; NMT, Neuromuscular training; RT, Regular training; CG, Control Group; QG, Quadriceps Strengthening Group; HG, HIP Strengthening Group; SG, Stretching Group; EP, Exercise Program Group; RM, Repetition maximum.

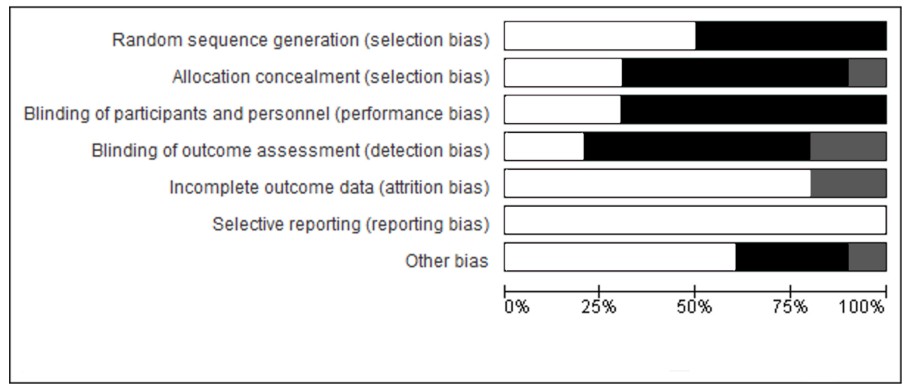

**Figure 2** **Risk of bias of the included studies.** The black fill indicates high risk of bias. The grey fill indicates unclear risk of bias. The white fill indicates low risk of bias.

## Quality assessment

The methodological quality of the included studies was evaluated based on the Cochrane risk bias assessment tool. The tool assessed random sequence generation, allocation concealment, blinding of participants and personnel, blinding of outcome assessment, incomplete outcome data, selective reporting, and other biases in the studies. Items were scored as "yes", "no", or "unclear". The details of the analysis of the risk of bias are described in Fig. 2.

## RESULTS

### Characteristics of included studies

A total of 590 articles were identified in the database search, but only two were included in the final review (*Bell et al., 2013*; *Barendrecht et al., 2011*). Additionally, thirteen other potential papers were identified when manually checking the reference lists of the included articles. Only eight fulfilled the inclusion criteria (*Sheerin, Hume & Whatman, 2012*; *Baldon et al., 2014*; *Czasche et al., 2017*; *Araújo et al., 2017*; *McCurdy et al., 2012*; *Jeong, Choi & Shin, 2020*; *Thompson-Kolesar et al., 2017*; *Saad et al., 2018*). Therefore, a total of ten studies were included in the final review.

In general, the results of included studies are differed based on the types of exercise prescribed in the training program and the participants involved. Most studies focused on strengthening the hamstring and quadriceps muscles. Only *Bell et al. (2013)* targeted the improvement of the hip and ankle muscles' strength and flexibility. The duration and frequency of exercise training program ranged from three to 12 weeks, with two to three sessions per week. The participants in the included studies were volunteers (*Bell et al., 2013*), university students (*Araújo et al., 2017*), handball players (*Barendrecht et al., 2011*), young athletes (*Sheerin, Hume & Whatman, 2012*), recreational athletes (*Czasche et al., 2017*; *Jeong, Choi & Shin, 2020*), preadolescent and adolescent soccer athletes (*Thompson-Kolesar et al., 2017*), females with previous athletic experience (*McCurdy et al., 2012*) and patients with patellofemoral pain syndrome, PFPS (*Baldon et al., 2014*; *Saad*

*et al., 2018*). Six studies involved only women (*Baldon et al., 2014*; *Saad et al., 2018*; *Araújo et al., 2017*; *Czasche et al., 2017*; *McCurdy et al., 2012*; *Thompson-Kolesar et al., 2017*), one study included only men (*Jeong, Choi & Shin, 2020*) and the remaining studies included both genders.

Regarding the study procedures, *Czasche et al. (2017)*, *Jeong, Choi & Shin (2020)* & *McCurdy et al. (2012)* did not mention any screening procedure used to evaluate excessive DKV. Moreover, *Baldon et al. (2014)* and *Saad et al. (2018)* did not specify any screening procedure to evaluate excessive DKV because they implied the presence of knee valgus based on the diagnosis of PFPS. However, other five studies, namely *Barendrecht et al. (2011)*; *Bell et al. (2013)*; *Araújo et al. (2017)*; *Sheerin, Hume & Whatman (2012)*; *Thompson-Kolesar et al. (2017)*, screened the participants for excessive DKV through various types of test. *Barendrecht et al. (2011)* measured knee valgus from the drop-jump test by evaluating the knee minimum distance values (i.e., less than 49.96% showed the presence of knee valgus) using 2-D analysis. Meanwhile, *Bell et al. (2013)* visually confirm the presence of knee valgus during double-legged squat whereas, *Araújo et al. (2017)* screened DKV during 18-cm step-down task. *Sheerin, Hume & Whatman (2012)* screened DKV based on the pre-running test's knee abduction angle while, *Thompson-Kolesar et al. (2017)* evaluated baseline knee valgus angle and moments for double, single-legged landing, preplanned and unanticipated cutting tasks.

The effectiveness of an exercise intervention program in improving DKV was evaluated using a variety of tasks such as step-up and step-down (*Saad et al., 2018*; *Araújo et al., 2017*), DLS (*Bell et al., 2013*), single-leg squat (*Baldon et al., 2014*), double, single-legged landing, preplanned and unanticipated cutting (*Thompson-Kolesar et al., 2017*), side step cutting (*Jeong, Choi & Shin, 2020*), drop jump (*Barendrecht et al., 2011*), unilateral and bilateral landings (*Czasche et al., 2017*), unilateral and bilateral drop jumps (*McCurdy et al., 2012*), and running analysis (*Sheerin, Hume & Whatman, 2012*). Our results showed Seven studies reported significant difference in DKV following the exercise intervention program (*Saad et al., 2018*; *Bell et al., 2013*; *Baldon et al., 2014*; *Jeong, Choi & Shin, 2020*; *Thompson-Kolesar et al., 2017*; *Barendrecht et al., 2011*; *Sheerin, Hume & Whatman, 2012*). On the other hand, *Czasche et al. (2017)*, *Araújo et al. (2017)* and *McCurdy et al. (2012)* did not detect any significant improvement on DKV after the exercise intervention program.

## DISCUSSION

Previous systematic reviews revealed the association of increased ankle dorsiflexion ROM (i.e., bottom-up kinetic chain) and hip strength (i.e., top-down kinetic chain) in reducing DKV (*Lima et al., 2018*; *Dix et al., 2018*). Additionally, a narrative review found that neuromuscular exercises targeting hip musculature strength may alter DKV (*Ford et al., 2015*). However, to date, there is no systematic review investigating the impact of exercise intervention based on top-down and bottom-up kinetic chains on DKV. We found that several exercise programs have successfully minimized excessive DKV in several tasks, for example, step-up and step-down (*Saad et al., 2018*), double-legged squat (*Bell et al., 2013*), single leg squat (*Baldon et al., 2014*), double-legged landing (*Thompson-Kolesar et*

*al., 2017*), drop jump (*Barendrecht et al., 2011*), side step cutting (*Jeong, Choi & Shin, 2020*) and running gait (*Sheerin, Hume & Whatman, 2012*). However, some studies showed no changes on DKV particularly during unilateral and bilateral landings (*Czasche et al., 2017*), unilateral and bilateral drop jumps (*McCurdy et al., 2012*) and step-down task (*Araújo et al., 2017*) following exercise intervention programs.

*Sheerin, Hume & Whatman (2012)* observed that eight weeks of lower limb functional exercise program minimally altered the knee frontal plane motions during running. They randomly assigned 19 youth athletes from a long-term athletic development program to either the control group that received only upper limb strengthening exercises or the experimental group that received both upper limb strengthening and lower limb functional exercises that focused on minimizing knee valgus angle. The experimental group performed functional weight bearing exercises aimed at reducing knee valgus angle as well as open and closed kinetic chain exercises to promote hip muscle activation. Thus, the exercise training involved a top-down kinetic chain that emphasized hip muscle to reduce knee valgus. The study was among the first to describe the effects of a lower limb functional exercise intervention on the running mechanics in the cohort of youth athletes. The minimal changes of knee valgus angle could be due to the studied population (i.e., youth athletes) who are accustomed to regular high-intensity training. Additionally, the study provided important insight on injury prevention as injury rates were often high among youth athletes (*Mail et al., 2019*).

*Barendrecht et al. (2011)* showed that neuromuscular training improved contact time, knee valgus and knee flexion angles during a drop jump test. They divided 80 adolescent handball players into two groups that received either neuromuscular training (NMT) ($n = 49$) or regular training (RT) ($n = 31$). The exercises in the NMT group included a combination of top-down and bottom-up kinetic chain exercises that were focused on enhancing hamstrings muscular strength and ankle motion control to alter knee mechanics. Meanwhile, the RT group received only regular handball training. Following that, the participants were also grouped based on their knee minimum distance values measured during landing or take-off in a drop jump test (*Noyes et al., 2005*). By using a cut-off point of 49.96%, those with knee minimum distance below the cut-off point value were assigned to the below-average valgus aligned (BAVA) group as compared to those in the above-average valgus aligned (AAVA) group (*Barendrecht et al., 2011*). The integrated NMT group displayed better knee kinematics and single-leg stability than RT, particularly among adolescent handball players with AAVA. Thus, it is recommended to incorporate the NMT program to improve knee valgus angle particularly among adolescent handball players who were predisposed to a high risk of ACL injury (*Griffin et al., 2006*). In fact, the overall rate of injury among this group of players was comparable to those in the senior handball team (*Olsen et al., 2006*).

*Thompson-Kolesar et al. (2017)* investigated the effects of FIFA Medical and Research Center (F-MARC) 11+ injury prevention warm-up program on biomechanical risk factors for ACL injury among pre-adolescent and adolescent female soccer players. Each age group was divided into two groups; intervention (received F-MARC 11+ injury prevention warm-up program in 15 sessions for 7–8 weeks (2 sessions per week)) and control (did

not received any program). The F-MARC 11+ involved exercises for core strength, neuromuscular control and balance, eccentric training of the hamstrings, plyometric and agility. The program involved a combination of top-down and bottom-up kinetic chain to alter the mechanisms of knee valgus formation. Then, the biomechanical changes were evaluated during four functional tasks; double-legged landing, single-legged landing, pre-planned and unanticipated cutting. The authors noted that the preadolescent athletes improved more in knee valgus angle and moment during the double-legged jump than adolescent athletes, implying that they can gain more from participation in the F-MARC 11+ program.

Bell et al. (2013) recruited a total of 32 volunteers with knee valgus before assigning them randomly to either the control or intervention group. Knee valgus was identified visually through the usage of corrected heel lift during a double-legged squat (DLS). Bell, Padua & Clark (2008) mentioned that corrected heel lift during DLS might restrict certain ankle motion, especially ankle dorsiflexion associated with the bottom-up mechanism of DKV. If the midpoint of the patella moved medially to the great toe in three to five squats, the participants were identified as having knee valgus. The intervention program was comprised of hip- and ankle-focused muscle strengthening and ankle flexibility exercises for two to three weeks. Therefore, the intervention program used a combination of top-down and bottom-up kinetic chain to reduce knee valgus. As for the control group, they did not receive any exercise program throughout the intervention period. The intervention group showed a 64% reduction of knee valgus during squat. The ankle strength and flexibility (i.e., bottom-up kinetic chain) may alter the individual's poor movement patterns by limiting medial knee displacement (MKD) and increasing the ankle dorsiflexion ROM during DLS (Bell et al., 2013). This is due to the ankle being placed in a more plantarflexed position, which reduced the knee abduction. The study by Bell et al. (2013) was among the first to demonstrate changes in the knee valgus angle during DLS following bottom-up DKV kinetic chain (i.e., ankle-focused) exercise intervention in individuals with identified knee valgus. However, it is crucial to highlight that the physical activity level of the participants in this study was not assessed or controlled, which may have influenced the results. Moreover, fitness level and gender of participants may influence the outcomes of intervention. For instance, in a study by Mohd Azhar et al. (2019), no association was observed between foot positions and lower limb kinematics during single leg squat (i.e., bottom-up kinetic chain) among highly trained male youth athletes, although Ishida et al. (2014) found significant association of similar variables among sedentary females.

Baldon et al. (2014) found that functional stabilization training (FST) consisted of hip muscle strengthening, lower limb and trunk movement control exercises (i.e., a combination of top-down and bottom-up kinetic chain exercises), modified the knee kinematics in the frontal plane during single leg squat (SLS). In this study, a total of 31 female recreational athletes with PFPS were assigned randomly to either standard training (ST) or FST group. The ST group received stretching, traditional weight-bearing and non-weight-bearing exercises that stressed on strengthening the quadriceps (i.e., top-down kinetic chain only). Although no DKV screening test was conducted by Baldon et al. (2014), PFPS was regularly associated with excessive DKV (Hewett et al.,

*2005*). The study highlighted the contribution of FST towards improving the lower limb control on the frontal plane motion following eight weeks of intervention. Nevertheless, this findings were only limited to females with PFPS (*Baldon et al., 2014*).

*Jeong, Choi & Shin (2020)* reported that the intervention group showed reduced knee valgus during side-step cutting task following ten weeks of core strength training. This study involved 48 male participants who were assigned randomly to either the intervention or control group. The intervention group performed core strengthening exercises (i.e., top-down kinetic chain) for ten weeks with three sessions per week while the control group continued their daily routine. The results corroborate the findings of *Willson, Ireland & Davis (2006)*, who discovered that participants with greater isometric core strength had reduced knee valgus angle during SLS. *Jeong, Choi & Shin (2020)* concluded that core strengthening, which involves top-down kinetic chain exercises, may assist male athletes in avoiding knee valgus alignment during side-step cutting.

*Saad et al. (2018)* reported that eight weeks of hip and quadriceps exercises (i.e., top-down kinetic chain) significantly reduced knee valgus angle during step tasks. A total of 40 female recreational athletes with PFPS were randomly assigned to hip-focused exercises, quadriceps-focused exercises, stretching exercises, and control (i.e., no intervention) groups. DKV was evaluated using a 3-D analysis of tibiofemoral flexion at 45° during step tasks. The step-up task was associated with a higher DKV incidence rate because it is a challenging motion for individuals with PFPS (*Saad et al., 2011*). From the pre-test results, the valgus movement pattern appeared for 87.18% during step-up and 82.05% during step-down. However, these values decreased at the end of the intervention program (i.e., hip and quadriceps exercises) to 66.67% during step-up and 48.72% during step-down.

Three other included studies found no significant changes in DKV following exercise intervention programs. *Czasche et al. (2017)* reported a change in the medial/lateral loading of the knee following eight weeks of lower limb strength training (i.e., top-down kinetic chain) despite a small non-statistically significant difference in the kinematics and GRF during drop landing. The study involved 16 untrained and healthy young students assigned to either control or intervention group. The intervention group received a leg strengthening exercise program for eight weeks while the control group continued their daily routine. No change in DKV could be due to the period of neural adaptation for strength training among untrained individuals. Previous research suggests that the knee valgus motion may be altered in highly untrained subjects who make significant gains in strength. For instance, *Mascal, Landel & Powers (2003)* observed reduced knee valgus motion during step-down task in two untrained patients with PFPS. The patient's strength increased by 50–110% in specified muscle groups after resistance training, indicating major muscle weakness prior to training (*Mascal, Landel & Powers, 2003*). Thus, for untrained participants, longer duration of lower limb strengthening program may be needed for observable DKV kinematical changes.

*McCurdy et al. (2012)* found no significant difference in knee valgus angle during bilateral (60 cm) and unilateral (30 cm) drop jumps between groups after eight weeks of resistance training. Twenty-nine young adult females with previous athletic experience were randomly assigned into either control or resistance training group. For two sessions a

week, the experimental group engaged in progressive load of resistance training. Although multi joints weight-bearing training with free weights is assumed to be more sport-specific than resistance band training, it appears that neither form of training affect knee valgus angle during landing tasks (*McCurdy et al., 2012*). Even though they observed that 1RM squat increased by 19%, athletes with previous resistance training experience will need more strength gains to dramatically change frontal plane kinematics during a drop jump task. Hence, the authors propose that long-term resistance training with a focus on increasing frontal plane intensity might be needed to change frontal plane kinematics.

*Araújo et al. (2017)* reported no significant differences in transverse plane knee kinematics during step-down task among women with DKV. The five exercises in the experimental group comprised of hip and trunk (i.e., top-down kinetic chain) muscle strengthening exercises for three sessions per week. The load of the exercises was increased gradually over the eight-week intervention. Meanwhile, the control group did not receive any training program. The step-down task was used to evaluate the presence of DKV among the participants, which was confirmed when the tibial tuberosity shifted over the second toe on a vertical imaginary line. Despite the high training volume, the intervention was not adequate to alter any frontal and transverse kinematic variables. This may be caused by the lack of specificity of the prescribed exercises to the tested motions. Thus, they suggested a longer period of interventions and specific exercises to strengthen certain muscles such as hamstring and gastrocnemius, to produce more significant changes on the knee kinematics. They also addressed the need for additional functional training, such as real-time feedback training during the step-down task to help the participants inspect the changes in hip function. Despite a review from *Dix et al. (2018)* that showed the relationship between hip muscle strength and DKV, *Araújo et al. (2017)* did not report any significant improvement in DKV following eight-weeks of intervention. To the best of our knowledge, this study represented one of the negative findings about top-down kinetic chain exercises on DKV.

About half of the included studies showed high risk of bias in terms of randomization, allocation concealment, blinding of participants, study personnel and outcome assessors, which could have influenced their results. Also, the review was limited to the knee biomechanical outcomes thus, future studies should consider other factors such as hip and ankle biomechanics.

## CONCLUSION

In this review, we investigated the effects of exercise interventions, based on either top-down or bottom-up kinetic chain, on minimizing DKV during specific tasks. As DKV is one of the main factors that could increase the risk of lower limb injury, findings from this review highlighted the importance of prescribing exercise training program to improve DKV.

Four studies (*Barendrecht et al., 2011*; *Bell et al., 2013*; *Baldon et al., 2014*; *Thompson-Kolesar et al., 2017*) found significant changes in knee valgus following a training program that combined both top-down and bottom-up kinetic chain exercises. Three studies (*Sheerin, Hume & Whatman, 2012*; *Saad et al., 2018*; *Jeong, Choi & Shin, 2020*) discovered that top-down kinetic chain exercise intervention programs were also effective in reducing

knee valgus. However, *McCurdy et al. (2012)*, *Czasche et al. (2017)* and *Araújo et al. (2017)* did not detect any significant effects of the exercise intervention in minimizing DKV. Therefore, we concluded that a combination of hip- and ankle- focused exercises and hip-focused (top-down kinetic chain) exercises only are more favorable than exercise program consisted of ankle-focused (bottom-up kinetic chain) exercises only. Apart from focusing on the load and volume of prescribed exercises, the specificity of exercises should also be emphasized. These findings might help the athletes and coaches to design appropriate exercise programs in reducing DKV.

## ACKNOWLEDGEMENTS

We thank Li Shuoqi for his assistance in preparing the figures for the manuscript.

### Funding
This work was supported by the Ministry of Higher Education Malaysia (FRGS/1/2020/SKK06/USM/03/10). The funders had no role in study design, data collection and analysis, decision to publish, or preparation of the manuscript.

### Grant Disclosures
The following grant information was disclosed by the authors:
The Ministry of Higher Education Malaysia: FRGS/1/2020/SKK06/USM/03/10.

### Competing Interests
The authors declare there are no competing interests.

### Author Contributions
- Farhah Nadhirah Aiman Sahabuddin conceived and designed the experiments, performed the experiments, analyzed the data, prepared figures and/or tables, authored or reviewed drafts of the paper, and approved the final draft.
- Nazatul Izzati Jamaludin conceived and designed the experiments, performed the experiments, analyzed the data, authored or reviewed drafts of the paper, and approved the final draft.
- Nurul Hidayah Amir analyzed the data, authored or reviewed drafts of the paper, and approved the final draft.
- Shazlin Shaharudin conceived and designed the experiments, analyzed the data, prepared figures and/or tables, authored or reviewed drafts of the paper, and approved the final draft.

### Human Ethics
The following information was supplied relating to ethical approvals (i.e., approving body and any reference numbers):

Universiti Sains Malaysia granted ethical approval to carry out the study within its facilities (Ethical Application Ref: USM/JEPeM/18070316).

## Data Availability

The raw data is available as a summary of included papers (Table 1), keywords used for article search in the Methods, and as a PRISMA flowchart (Fig. 1).

## Supplemental Information

Supplemental information for this article can be found online at http://dx.doi.org/10.7717/peerj.11731#supplemental-information.

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
