# Peer review of "The effects of hip- and ankle-focused exercise intervention on dynamic knee valgus: a systematic review"

_PeerJ, doi:10.7717/peerj.11731_

## Round 0.1 · original submission · Major Revisions

Both reviewers mentioned that the motivation for the review needs to be strengthened. It should be clear what was missing in previous reviews, such that a new review is needed. One motivation seems to be the top-down and bottom-up mechanisms of dynamic knee valgus. This was not sufficiently convincing for reviewer 1, and both reviewers mentioned that this topic should play a major role in the discussion. The conclusions should be strengthened, and be more specific. "Exercise intervention" includes a broad range of interventions, and it would be helpful to identify the interventions that appear most promising.

·

Basic reporting

- The introduction shows context and literature is well referenced.
- The language is in many circumstances unclear and ambiguous and needs improvement.
- The article is well structured. The table needs proper formatting.
- The methods are mostly described with sufficient detail.

Experimental design

- 1) The research question is too loosely defined, including all types of exercise interventions (strength training, balance training, explosive training…), participants (age, sex, sport, training experience, with and without knee pain), measurement methods for assessing DKV (visual inspection, motion capture, 2D analysis…), and testing tasks (step-up, single-leg squat, drop jump, gait…), making it difficult to produce valid results and draw meaningful conclusions. In addition, the outcome measures are not well defined, including both knee kinematics, knee valgus angle, knee abduction moment…
- 2) It is not clear why this review is needed and how it differs from previous systematic reviews assessing the effect of training interventions on DKV (e.g. https://www.ncbi.nlm.nih.gov/pmc/articles/PMC6604048/ ). Also, the necessity of focusing on top-down or bottom-up kinetic chains is not clear and not followed up upon in the discussion. It is not clear how this review fills an identified knowledge gap.

Validity of the findings

- Poor synthesis of study results and respective recommendations.
- The discussion is mostly a summary of the identified studies and their limitations.
- Conclusions are not well stated and not limited to supporting results.

Additional comments

- I read this manuscript with great interest and I commend the authors for their extensive work. The topic is of great importance for everyone involved in preventive training of knee injuries. However, this systematic review has important methodological limitations as highlighted above.

·

Basic reporting

I am assuming Table 1 in it's current format is not the final version of what will be published. It's current form is stretched over multiple pages, with improper formatting on the column width, which splits words, has the text centered incorrectly etc. I believe this would be fixed after the paper is accepted, but please double check this.

Experimental design

My two big, overall methods questions are:

1) I think some context here as to why “knee abduction” was not used as a search term is warranted. This could just be a comment back to the reviewer. But I would like to know why this wasn’t used, and if you believe this could impact the implications of your systematic review?

2) Do you think it is appropriate to include a study (Horsak et al. 2019) that focused on obese children as part of the 8 studies included in the review? The reason I ask this is because obesity is a special population where the obesity component is one of the primary drivers behind DKV. Comparing interventions with obese children to the other 7 studies that feature average-weight adults that are recreationally active might not be an appropriate comparison.

Validity of the findings

An overall comment for both the discussion and conclusion: In L94 – 96 of the Introduction, the point is made that there has been no systematic review that has focused on exercise interventions from a top-down or bottom-up perspective. In the discussion, the 8 studies used in the review are identified as top-down or bottom-up. However, there is no discussion by the authors on their perspective of if a top-down or bottom-up approach is more appropriate when using exercise interventions to improve DKV. Since this emphasis was made in the Introduction, a few sentences to acknowledge and discuss this should be added to the Discussion or Conclusion section. This would greatly strengthen the systematic review.

---

## Round 0.2 · Major Revisions

Dear authors,

I did not send your revised manuscript to the reviewers, because your rebuttal document did not include a point-by-point response to all of the reviewer comments. It seems that you only responded to the comments in the document that reviewer 2 sent. The reviewer comments from your initial submission are available in the online system, and also I have attached them here as a PDF. Please make sure you respond to all points, and then submit your revision.

---

## Round 0.3 · accepted · Accept

Thank you for the thorough revision and rebuttal.

·

Basic reporting

N/A

Experimental design

N/A

Validity of the findings

N/A

Additional comments

N/A